# Mechanisms of Male Reproductive Toxicity of Polybrominated Diphenyl Ethers

**DOI:** 10.3390/ijms232214229

**Published:** 2022-11-17

**Authors:** Olatunbosun Arowolo, J. Richard Pilsner, Oleg Sergeyev, Alexander Suvorov

**Affiliations:** 1Department of Environmental Health Sciences, University of Massachusetts, Amherst, MA 01003, USA; 2Department of Obstetrics and Gynecology, Wayne State University School of Medicine, Detroit, MI 48201, USA; 3Belozersky Institute of Physico-Chemical Biology, Lomonosov Moscow State University, 119992 Moscow, Russia

**Keywords:** PBDE, dose–response relationships, mTOR signaling, steroidogenesis, metabolic disruption, sperm epigenome

## Abstract

Polybrominated diphenyl ethers (PBDE) are a group of flame retardants used in a variety of artificial materials. Despite being phased out in most industrial countries, they remain in the environment and human tissues due to their persistence, lipophilicity, and bioaccumulation. Populational and experimental studies demonstrate the male reproductive toxicity of PBDEs including increased incidence of genital malformations (hypospadias and cryptorchidism), altered weight of testes and other reproductive tissues, altered testes histology and transcriptome, decreased sperm production and sperm quality, altered epigenetic regulation of developmental genes in spermatozoa, and altered secretion of reproductive hormones. A broad range of mechanistic hypotheses of PBDE reproductive toxicity has been suggested. Among these hypotheses, oxidative stress, the disruption of estrogenic signaling, and mitochondria disruption are affected by PBDE concentrations much higher than concentrations found in human tissues, making them unlikely links between exposures and adverse reproductive outcomes in the general population. Robust evidence suggests that at environmentally relevant doses, PBDEs and their metabolites may affect male reproductive health via mechanisms including AR antagonism and the disruption of a complex network of metabolic signaling.

## 1. Introduction

Polybrominated diphenyl ethers (PBDEs) are a group of halogenated biphenyl chemicals that structurally contain bromine atoms [1]. Depending on the number and location of the bromine atom(s) on aromatic rings, PBDEs are classified into 209 congeners [2]. This group of chemicals has been used extensively as flame retardants in various consumer products such as construction, textile, electrical, plastics, and furniture materials [3,4]. Globally, three major commercial mixtures of PBDEs have been produced and used: deca-, octa-, and penta-BDE [5]. Historically, the use of PBDE in various forms dates to the 1960s; it was used continuously for various purposes for almost five decades before most congeners were banned [6] (Suvorov and Takser, 2008). During the Stockholm Convention in 2009 and 2017, deca-BDE, octa-BDE, and penta-BDE were added to the list of persistent organic pollutants (POPs) that should be eliminated from the environment [7]. The highest demand and production of total PBDEs was documented to be in the year 2003 and reached approximately 90 kt per year [8]. Penta- and octa-BDE were voluntarily removed by industry from the USA market by 2004 [9], while deca-BDE was removed by 2013 [10]. Studies have shown that levels of PBDEs in the United States have been and remain higher than in any other country in the world [11].

PBDEs escape to the environment during their production, utilization, recycling, and disposal; this is due to their characteristic nature of not binding to complex polymers [12]. PBDEs are highly ubiquitous, persistent in nature, and can travel long distances [4,13,14]. The high lipophilicity of PBDEs (Log K_ow_ of 5.74–8.27) is responsible for their persistence, bioaccumulation, and biomagnification [15]. Humans are usually exposed to PBDEs via the inhalation and ingestion of dust containing PBDEs, and the ingestion of contaminated food [16]. The half-lives of PBDE congeners in the human body are estimated to be between 1 and 12 years [17,18]. The highest levels of PBDEs were detected in human adipose tissue samples collected in New York from 2003–2004 and ranged from 17 to 9630 ng/g lipid [19].

Although the wieldy uses of PBDEs were restricted in most developed countries, the high lipophilicity and stability of PBDEs in the environment and their long half-life in human tissues contribute to continuous exposures. For example, an epidemiological study of 1253 women in California suggests that serum PBDE levels continued to increase despite the phase-out [20]. Different health outcomes were found to be affected by PBDEs, with the impairment of male reproduction being among the most sensitive. Human and animal data indicate that PBDEs can affect multiple male reproductive outcomes, such as androgen levels in blood [21], sperm quality [22], and the incidence of genital developmental abnormalities [23]. However, the molecular mechanisms of male reproductive toxicity remain poorly understood [24].

Over years of research, many studies reported discrepant results pertinent to the male reproductive toxicities of PBDEs, and many mechanistic hypotheses of the male reproductive toxicity of PBDEs were suggested based on experimental research and population studies. Thus, today, there is significant uncertainty and discrepancies in the understanding of both major molecular cascades affected by PBDEs and the major reproductive health outcomes induced by PBDE toxicity. The goal of this study is to conduct a comprehensive review of existing human and animal studies to characterize the male reproductive toxicity of PBDEs, identify the most sensitive endpoints, and discuss the molecular mechanisms involved in the male reproductive toxicity of PBDEs. Following the central principle of toxicology credited to Paracelsus: “the dose makes the poison”, we identify plausible molecular mechanisms by which PBDEs cause male reproductive toxicity in the general population using the relevance of doses producing the disruption of these mechanisms as a central tool.

## 2. Results

### 2.1. Reproductive Health Studies in Humans

Nine epidemiological studies investigated the role of PBDEs on the human male reproductive system (Table 1). In a prospective cohort study, the effects of ten serum congeners of PBDE (BDE-17, 28, 47, 66, 85, 99, 100, 153, 154, and 183) were cross-sectionally accessed on semen quality in 468 adult men recruited from different counties of Texas and Michigan in the United States. Out of the 10 PBDE congeners studied for five PBDEs, 50% of the serum samples were below LOD, some PBDEs were associated with increased abnormal sperm morphology (BDE-28, 153), and BDE-28 was associated with reduced sperm motility [25].

In a Canadian study of 153 men aged 18–41 years living in Montreal, the hair BDE-47 concentration (median (IQR) 9.4 (4.0–18.0 ng/g) was negatively associated with sperm motility. The quality of the sperm chromatin was not affected by the studied congeners [28]. In Japan, the effect of PBDE was assessed in the sperm samples of 10 young men who were recruited from the Department of Urology at a Private University in Kawasaki. Four congeners of PBDE (BDE-47, 99, 100, and 153) out of the twenty-nine studied were detected in the serum samples. An inverse correlation was observed between the serum levels of BDE-153 (median 0.72 ng/g lipids) and sperm concentration and testis size (r = −0.841, *p* = 0.002 and r = −0.764, *p* = 0.01, respectively) [26].

The effects of PBDE on semen quality were assessed in 32 adult men between the ages of 20 and 50 years in a rural community close to an electronic waste recycling area in Qingyuan, China. The sperm parameters were evaluated in relation to concentrations of a range of PBDE congeners in respective house dust and sperm samples. The concentrations of congeners were positively correlated with the dust (median BDE-28 1.46 ng/g, BDE-47 12.0 ng/g, and BDE-153 14.6 ng/g) and paired semen samples (median BDE-28 5.02 pg/g, BDE-47 6.75 pg/g, and BDE-153 7.36 pg/g). Semen BDE-47 was negatively associated with sperm concentration and total sperm count, while the dust levels of BDE-100 were negatively associated with sperm progressive motility and viability [30].

An association between the blood levels of PBDE and male reproductive parameters was studied in men from Ukraine, Poland, and Greenland. Blood and sperm samples from 100 men in each of the countries (total 299) were randomly selected. The effects of BDE-47 and BDE-153 on the men’s reproductive functions were studied using an adjusted linear regression model. BDE-153 and BDE-47 were detected in almost all serum samples, with the highest level in Greenland (median (IQR) for BDE-47 2.0 (0.6–6.9 ng/g lipids) and for BDE-154 2.7 (1.3–7.8 ng/g lipids)), but they were not associated with markers of semen quality and serum reproductive hormone [27].

Studies from a Canadian group reported an association between PBDE concentrations in maternal hair and the incidence of developmental genital malformations, cryptorchidism, and hypospadias in their sons. In a Montreal case-control study, eight congeners of PBDE (BDE-209, 183, 154, 153, 100, 99, 47, and 28) were assessed in the hair of 137 mothers of infants with cryptorchidism and 158 mothers of male infants who did not suffer from cryptorchidism. The concentrations of three PBDEs (BDE-99, 100, and 154) were significantly higher in the cases than in the controls. Double the risk of infants developing cryptorchidism was observed with every ten times increase in the concentration of BDE-99, BDE-100, or BDE-154 in the maternal hair [29]. Similarly, higher levels of total PBDE and five individual congeners (BDE-28, 47, 99, 153, and 154) were found in the hair of mothers whose infants had hypospadias (n = 152) than among controls (n = 64) in Toronto [31,32]. In a multivariable model, hypospadias was associated with a relative 48.2% (95% CI, 23.3–65.4%) higher maternal hair level of total PBDE including eight congeners (Poon et al., 2018). These findings are not supported by the study conducted in California, although the sample size was small in this study [33]. Maternal mid-pregnancy levels of individual PBDE congeners (BDE-28, 47, 99, 100, 153) were not associated with the risk of hypospadias (n = 20 vs. n = 28 control).

In the studies discussed above, PBDEs have been shown to have effects in the general population on various reproductive outcomes in humans, including sperm motility (BDE-28, 47, 100), sperm concentration (BDE-47, 153), total sperm count (BDE-47), sperm morphology (BDE-28, 153), cryptorchidism (BDE-99, 100, 154), hypospadias (total sum of eight PBDEs), and testes size (BDE-153) (Figure 1).

### 2.2. Reproductive Health Outcomes in Animal Studies

Several studies used animals to study the effects of PBDE on the male reproductive systems. Table 2 highlights the details of these studies. The developmental and adult male reproductive effects were examined separately, as in the early steps of development, xenobiotics may induce changes in morphogenic trajectories resulting in health outcomes very different from those of adult exposures.

#### 2.2.1. Developmental Effects

Many studies focused on the developmental toxicity of BDE-47, the dominating congener in human tissues, using different exposure windows in a rat model. In perinatally exposed rats, many male reproductive outcomes are impaired even if they are assessed many days after exposure cessation. For example, in one study, female rats were exposed to 0.1, 1, and 10 mg/kg/body weight (BW)/day of BDE-47 starting 10 days before mating till lactation cessation on postpartum day 21 [37]. On postnatal day (PND) 88, the pups’ relative testis weight was significantly reduced in rats exposed to all doses compared with the controls. The sperm motility parameters and sperm counts were also significantly reduced in the offspring of dams exposed to 10 mg/kg. In another study, pregnant rats were exposed to 0.2 mg/kg/BW/day of BDE-47 from gestational day (GD) 8 to PND21 [39]. On PND120, the absolute weight of testes, daily sperm production, and percentage of motile sperm were all reduced in the offspring of the exposed dams. In addition, the percentage of morphologically abnormal spermatozoa was higher in the offspring of exposed dams than in the control. This study also reported significant changes in testes transcriptome, including the suppression of genes essential for spermatogenesis and the activation of immune response genes.

Using the same model, the researchers also demonstrated the developmental effects of BDE-47 on sperm epigenome. First, they showed that perinatal exposure results in an increase in the DNA methylation of epididymal sperm in genes, promoters, and intergenic regions in younger rats (PND65), while the methylation of the same elements decreases in older exposed animals (PND120) [34]. In total, 21 and 9 exposure-related differentially methylated regions (DMRs) were identified in the sperm collected on PND65 and PND120, respectively, with two DMRs overlapping between the two time points. Further, the authors focused on age-dependent changes in the sperm epigenome, namely, profiles of DNA methylation [36] and small non-coding RNA (sncRNA) [35]. They demonstrated that both profiles change significantly with age: 5319 age-dependent DMRs and 1384 sncRNA with age-dependent expression were identified. Perinatal BDE-47 modified the normal dynamics of age-dependent changes in both DNA methylation and sncRNA in sperm. For both epigenetic markers, these changes may be interpreted as an acceleration of age-dependent changes in younger animals and their deceleration in older animals [35,36]. Genes associated with age-dependent DMRs as well as the gene targets of age-dependent sncRNA were highly enriched with categories relevant to embryonic development. Thus, it is plausible that the modification of epigenetic programs in sperm following developmental exposure to PBDE may result in the altered development of embryos conceived by exposed fathers.

Three studies used PBDEs other than BDE-47 to assess the pre-/perinatal male reproductive effects in rats. The perinatal exposure (GD6-PND18) of pregnant/lactating rats to 18 mg/kg/BW of a commercial penta-BDE mix, DE-71 (a mixture of BDE-47, 99, 100, 153, and 154), led to a 1.3-fold increase in pups’ testes weight on PND31 [41]. The in utero exposure to 40 mg/kg/BW DE-71 also resulted in decreased anogenital distance in male offspring, a marker of estrogenic or antiandrogenic effects of exposure [43]. In another study, the male offspring of pregnant rats were exposed to a single dose of BDE-99 at 0.06 or 0.3 mg/kg on GD6; however, no significant changes were observed in the absolute testes and epididymal weight, prostate, seminal vesicles, serum luteinizing hormone, or testosterone levels [42]. The testes of rats exposed to 0.3 mg/kg were considered small when expressed as a percentage of body weight. The daily sperm production, sperm, and spermatid counts were significantly reduced by both doses [42].

Few studies used prepubertal exposure window to study the developmental effects of PBDE in rat models. For example, 14 days of exposure during prepuberty (PND21–PND35) to 0.1, 0.2, and 0.4 mg/kg BDE-47 did not have any effect on the weight of the rats’ testes [38]. An increase in the numbers of Leydig cells, serum testosterone, and decreased serum luteinizing hormone levels were observed upon exposing rats to 0.4 mg/kg/BW. Another study used a prepubertal exposure window (PND23–53) to address the effects of DE-71 in rats. In animals exposed to 3 or 30 mg/kg/BW of DE-71, androgen-dependent tissues (ventral prostate and seminal vesicle), testes, and epididymal weight were not affected. However, 30 mg/kg caused a significant delay in preputial separation by 1.7 days [40].

Two mouse studies investigated the postnatal effects of BDE-209 on male reproductive function. In mice exposed to 10 mg/kg of BDE-209 from PND21 till PND71, no significant difference was observed in the testes, epididymal weight, seminal vesicles, sperm chromatin structure, motility, count, and testes morphology of the exposed group and the control [45]. Conversely, neonatal (PND1–5) exposures to much lower subcutaneous doses (0.025, 0.25, and 2.5 mg/kg/BW/day) resulted in the decreased testicular weight (0.025 and 0.25 mg/kg), epididymal weight (0.25 mg/kg), sperm counts (0.025 mg/kg), elongated spermatid (0.025 mg/kg) and Sertoli cell numbers (0.25 mg/kg) [46]. In addition, a significant reduction in the serum testosterone level was observed after exposure to 0.025 and 0.25 mg/kg of BDE-209 [47].

One recent study analyzed the effects of prenatal (GD1–GD21) exposure to BDE-99 on reproductive development in mice [44]. Doses as low as 0.2 mg/kg/BW decreased the anogenital distance in male offspring, decreased testes size and Leydig cell numbers, increased the incidence of hypospadias, decreased blood testosterone, and resulted in the altered expression of steroidogenic enzymes at the gene and protein levels [44].

Thus, mouse and rat studies showed that perinatal and developmental postnatal exposures to individual PBDE congeners at low, environmentally relevant doses (as low as 0.1 mg/kg/BW/day for BDE-47 and 0.025 mg/kg/BW/day for BDE-209) have morphogenic effects on the male reproductive system, resulting in smaller testis weight and abnormal outcomes of spermatogenesis [37,39,46]. The peripubertal window may be less sensitive to PBDE reproductive toxicity [40,45]; however, more studies are needed in this developmental window. Although several studies demonstrate reduced testes weight following developmental exposures to PBDE, one shows the opposite change [41], suggesting that the direction of change of male reproductive outcomes may depend on the dose, congener, and/or developmental window.

#### 2.2.2. Adult Effects

We identified five studies that tested the male reproductive toxicity of PBDE in mice and four studies using rat models.

Two mouse studies focused on BDE-47. In one study, 30 days of exposure of adult animals to 0.0015, 0.045, or 30 mg/kg of BDE-47 led to a decrease in the rate of sperm capacitation in all exposed groups [53]. No significant difference between the sperm morphology of exposed and control groups was observed in this study; however, some sperm motility parameters were impaired in some exposed groups. In another BDE-47 study, the histopathological examination of mice exposed to 1.5, 10, and 30 mg/kg of BDE-47 for six weeks showed that sperm decreased in the epididymal lumen at 10 and 30 mg/kg [54].

Other mouse studies analyzed the reproductive effects of BDE-3 and BDE-209. The exposure to 0.0015, 1.5, 10, and 30 mg/kg BDE-3 for six consecutive weeks resulted in decreased sperm counts in mice exposed to 1.5 mg/kg or higher doses, decreased germ cells in seminiferous tubules, and also decreased mature spermatozoa in the epididymis of animals exposed to 30 mg/kg BDE-3 [52]. The exposure of adult mice to 7.5, 25, or 75 mg/kg/BW/day of BDE-209 also resulted in significantly reduced sperm numbers, while only 25 and 75 mg/kg significantly reduced sperm motility [55]. The histopathological examinations also showed a dose-dependent significant reduction in the height of germinal epithelium for the three doses [55]. In another study, adult mice were exposed for 8 weeks to 20, 100, or 500 mg/kg BDE-209 [56]. Changes in the weight of reproductive organs, testes histology, concentrations of reproductive hormones, and sperm morphology were only observed at 100 and 500 mg/kg BDE-209 in this study [56].

Two rat studies explored the effects of BDE-47 in adults. Exposures to 0.03 and 1 mg/kg BW/day BDE-47 significantly decreased the serum testosterone levels and increased the number of multinucleated giant cells in the testes that arose from spermatocytes that aborted meiosis, while the daily sperm production was decreased in a group exposed to 1 mg/kg BW/day BDE-47 [48]. In another study, 0.03 and 20 mg/kg BDE-47 also reduced the serum testosterone levels and altered the cellular organization of the seminiferous epithelium. In addition, 20 mg/kg BDE-47 also aborted meiosis in the spermatocytes [49].

Studies using adult rat models also assessed the male reproductive toxicity of BDE-209 and DE-71. In the study with BDE-209, adult male rats were exposed to a range of doses ranging from 1.87 to 60 mg/kg/BW/day for 28 days, and a benchmark dose approach was used to identify the dose that results in a 10% change in the reproductive outcome [51]. For seminal vesicle/coagulation gland weight (most sensitive outcome), a 10% decrease was achieved at the 0.2 mg/kg BW/day exposure level. A dose-dependent decrease of the epididymal weight was observed as well, although the benchmark dose was not calculated for this outcome. No changes in sperm counts or morphology were observed in this study [51]. In adult male rats exposed to 3 and 30 mg/kg DE-71, a decrease in serum androstenedione (3 and 30 mg/kg) and a significant decrease in the ventral prostate (30 mg/kg) were observed, while the serum testosterone (3 mg/kg), LH, and estrone (3 and 30 mg/kg) levels were all increased in the exposed group [50].

Few field studies demonstrated the effects of PBDEs in large mammals. For example, the testicular levels of PBDEs negatively correlated with Sertoli cell numbers and germ cells’ proliferative activity in dogs residing in Finland, Denmark, and the UK [57]. Testis size was significantly negatively associated with subcutaneous levels of PBDE in 20 subadult polar bears in Greenland [58].

Rodent laboratory studies as well as wildlife studies demonstrate that PBDE exposure is a potent factor affecting male reproductive system physiology in adult age.

### 2.3. Most Sensitive Outcomes

The most sensitive male reproductive outcomes identified in developmental and adult animal experiments are as follows (Figure 1). Among the developmental studies, the lowest dose capable of producing health effects was reported in the study where mice were exposed to 0.025 mg/kg/BW/day of BDE-209 for five days following birth [46]. This dose of BDE-209 decreased the testicular weight and the sperm and elongated spermatid count. The subcutaneous delivery of BDE-209 used in this study has low relevance for human exposure, as humans are mostly exposed to PBDE via inhalation and ingestion. The findings of this study keep their relevance for human health, however, as the difference in the routes of exposure likely does not have a significant effect on the resulting body burdens, as the rate of absorption after the oral administration of PBDEs is 75–85% in rodents [59,60,61].

Among adult studies, the lowest dose that produced changes in male reproductive outcomes was 0.0015 mg/kg of BDE-47 per day over 30 days [53]. That dose resulted in decreased sperm motility and sperm capacitation in rats. Thus, in adult animals, PBDEs may affect the physiology of spermatozoa at doses at which no effects on reproductive organ size or sperm production are seen.

In both developmental and adult rodent studies, the effects of PBDEs on the male reproductive system were seen at a microgram range of exposures, suggesting that in developed countries, almost 100% of males may experience changes in reproductive functions due to exposure to PBDE during their lifespan.

### 2.4. Mechanisms That Mediate Male Reproductive Toxicity of PBDE

Several different molecular and endocrine mechanisms that may be involved in the reproductive toxicity of PBDEs have been suggested. These mechanisms are reviewed below.

#### 2.4.1. Induction of Oxidative Stress

Oxidative stress is caused by the inability of the body to manage the imbalance in the cellular production and deactivation of reactive oxygen species (ROS) [62]. The oxygen-containing reactive species include superoxide radicals (O_2_^−^), hydroxyl radicals (OH), hydrogen peroxide (H_2_O_2_), peroxyl radical (LOO), and lipid hydroperoxides (LOOH) [63,64]. ROS are natural by-products of oxygen metabolism, and they participate in several physiological processes such as the immune response to pathogens, protein phosphorylation, and cellular signaling [62]. Spermatozoa are a source of ROS themselves and, additionally, ROS are produced in semen in large quantities by polymorphonuclear leucocytes [65,66,67,68].

The sperm membrane of spermatozoa has an unusual structure, which is enriched with lipids that are the main substrates for peroxidation: phospholipids, sterols, and saturated and polyunsaturated fatty acids [69,70,71]. This membrane structure makes spermatozoa particularly susceptible to damage by ROS [72,73,74,75]. Thus, excessive ROS can significantly affect sperm physiology through their effect on the sperm plasma membrane, resulting in decreased sperm motility and vitality [76,77,78,79], impaired capacitation [80], acrosome reaction [81], and other sperm parameters and outcomes of fertilization [82]. Although spermatozoa have an antioxidant defense system that detoxifies ROS, the overproduction of ROS can overwhelm these mechanisms and induce oxidative stress [83]. Overall, oxidative stress is one of the major causes of sperm damage involved in male infertility [83,84,85,86].

The exposure to xenobiotics including PBDEs can increase ROS production and induce oxidative stress in different tissues [87,88,89]. A growing body of studies using diverse biological systems demonstrate that PBDEs can affect ROS balance and induce changes to the antioxidant defense system. To name a few, the exposure of GC-2 cells to 8, and 32 μg/mL of BDE-209 led to an increase in ROS levels in the cell [55]. Metabolites of BDE-47 (3-MeO-BDE47, 3-MeO-BDE47, 5-MeO-BDE47, and 5-OH-BDE47) increased the activity of superoxide dismutase (SOD), decreased the levels of glutathione (GSH), and increased ROS in LO2 cells in a dose-dependent manner [90]. In another study, the exposure of LO2 cells to 10 and 50 μM BDE-209 also led to a significant increase in ROS levels [91]. The treatment of HS-68 human cell culture to 50 μmol/L BDE-47, 100 μmol/L BDE-99, and 2 μmol/L BDE-209 led to an increase in the levels of intracellular ROS [89]. In comparison with the control, a significant increase in the superoxide dismutase enzyme activity and a significant decrease in GSH reductase enzyme activity was observed in the erythrocytes of adult rats exposed to 0.6 and 1.2 mg/kg of BDE-99 [92].

Thus, it is not surprising that oxidative stress is considered a candidate mechanism that may connect PBDE exposure with adverse male reproductive outcomes. Several in vivo studies addressed this potential mechanism of reproductive toxicity mostly focusing on BDE-209. Testicular malondialdehyde (MDA), a marker of lipid peroxidation, was increased in mice testes exposed to 25 and 75 mg/kg of BDE-209, while SOD enzymatic activities were decreased [55]. In comparison with the control, SOD activity and GSH levels were also decreased in male mice dosed with 200 and 500 mg/kg/BW of BDE-209 [93]. MDA was also increased in adult mice testes exposed to 100 and 500 mg/kg/BW BDE-209 for 8 weeks, but not in mice exposed to 20 mg/kg [56]. In the same study, GSH levels were increased in the 20 and 500 mg/kg groups, although small n (4–5) and very high SD in the 20 mg/kg group may indicate a false positive finding. On the other hand, the increased expression of gene markers of endoplasmic reticulum stress (Atf6, Ire1) and gene (Bcl-2) and protein (Bax) markers of apoptosis in the 20 mg/kg group may indicate that this level and duration of exposure is sufficient to reach an oxidative stress response [56]. A significant increase in the sperm H_2_O_2_ was observed in mice exposed to 500 and 1500 mg/kg of BDE-209 [45]. In another study, mice were exposed to 10, 500, and 1500 mg/kg/BW of BDE-209 from GD 0 to 17 [94]. A significant increase in sperm H_2_O_2_ generation was observed at the lowest and highest doses in this study [94]. In lactating female mice orally gavaged with 500 and 700 mg/kg/BW of BDE-209, SOD and catalase enzyme activities were decreased in male pups on PND21 and PND28 from both exposure groups [95]. Moreover, the maternal exposure of female mice to 500 and 700 mg/kg of BDE-209 led to a decrease in the activities of catalase and SOD enzymes in the testes of the offspring. In the same study, an increase in the total ROS production was observed in the testes of offspring from all exposure groups [96].

All studies listed in the previous paragraph used very high doses of BDE-209 that are not relevant to human exposures. As we discussed earlier, the lowest doses of PBDEs that were shown to induce male reproductive toxicity in animal experiments were 0.025 mg/kg/BW/day in developmental and 0.0015 mg/kg BW/day in adult studies. These doses are three to six orders lower than most studies that have identified oxidative stress-related outcomes in male reproductive tissues. Therefore, these findings cannot be used to justify the role of oxidative stress in the male reproductive toxicity of PBDEs. One study tried to link male reproductive toxicity of low doses BDE-47 with oxidative stress mechanisms [48]. In this study, male reproductive outcomes and ROS levels in the seminiferous tubules were analyzed in rats exposed to 0.001, 0.03, and 1 mg/kg BW/day of BDE-47. The study reports reproductive toxicity at 0.03 mg/kg BW/day and the higher dose. However, changes in ROS were not significant at any dose, although these changes were claimed “dose-dependent” by the authors. In in vitro analysis, BDE-99 induced ROS in Leydig cells at a 10 uM concentration [44]. Given that blood concentrations of PBDE in the general population may only reach the nM range [97,98,99], it is unlikely that PBDE effects on Leydig cells found in some in vivo studies are due to an oxidative stress mechanism.

To conclude, although oxidative stress is a plausible mechanism that can be induced by PBDE and mediate reproductive toxicity, to date there is no reliable evidence to support this hypothesis.

#### 2.4.2. Metabolic Disruption

Recent research demonstrated that PBDEs are potent metabolic disruptors affecting lipid and glucose metabolism at environmentally relevant doses, although the mechanisms of this disruption are poorly understood. For example, serum triglycerides were significantly increased two-fold in mouse pups exposed to 0.2 mg/kg of BDE-47 perinatally [100]. Neonatal exposure to 1 mg/kg of BDE-47 produced the opposite effects on lipids in circulation: decreased blood triglycerides and increased liver triglycerides in adult mice [101]. In another study, perinatal exposure to 0.002 and 0.2 mg/kg BDE-47 caused a worsening of high-fat diet-induced obesity, hepatic steatosis, and impaired glucose homeostasis [102]. A recent mouse study reported the development of a diabetic phenotype in mice perinatally exposed to 0.1 mg/kg BW DE-71 [103]. These mice developed fasting hyperglycemia, glucose intolerance, and reduced insulin sensitivity. In rats exposed to 14 mg/kg BW/day of a mix of PBDE congeners, a decrease in insulin-stimulated glucose oxidation and an increase in isoproterenol-stimulated lipolysis was observed in adipocytes [104]. A significant disruption of the metabolism of lipids and carbohydrates was also observed in mice exposed to high doses of BDE-209 [91,95].

Similar evidence started to emerge recently from epidemiological studies. For example, among 34 obese Qatari individuals, insulin-resistant subjects had significantly higher levels of BDE-99, BDE-28, BDE-47, and the sum of penta-BDE in adipose tissue than insulin-sensitive counterparts [105]. Additionally, BDE-99 and BDE-28 positively correlated with fasting insulin levels in this study. An increased risk of type 2 diabetes (T2D) in association with dietary exposures to PBDE was found in a French study of 71,415 women including 3667 diagnosed with T2D, although the exposure assessment in this study was conducted using a scenario evaluation approach based on a questionnaire and may suffer from low accuracy [106]. Similarly, a positive association was found between total PBDEs and gestation diabetes mellitus (GDM) in Tehran women (70 cases vs. 70 controls) [107]. Another study on 147 mother–children pairs demonstrated that in utero BDE-99 was associated with lower childhood levels of triglycerides, high-density lipoprotein, and total lipids in children’s blood at 6–7 years of age [97], suggesting long-lasting metabolic effects of early-life exposure.

Overall, experimental and epidemiological evidence suggests that PBDEs may mimic and exacerbate the effects of a high-fat or high-calorie diet as they stimulate lipid accumulation in the liver and decrease glucose tolerance and insulin sensitivity. These symptoms resemble diabetic and obesity phenotypes, both contributing to male infertility via multiple mechanisms [108,109,110,111]. A discussion of these mechanisms is beyond the scope of the current review. However, it is important to mention that the metabolic disruption associated with obesity and/or diabetes results in a negative impact on semen parameters, including sperm concentration, motility, viability, and normal morphology [108,109,110,111].

Evidence that supports the hypothesis that PBDE reproductive toxicity is mediated via metabolic disruption was obtained recently in a metabolomics study where 76 differential metabolites were found in mouse testis tissue following exposure to 0.0015, 1.5, 10, or 30 mg/kg/BW of BDE-3 for six weeks [52]. These metabolites were enriched for nucleotide metabolism and lipid metabolism as well as pathways involved in the metabolism of several amino acids and riboflavin, an important player in carbohydrate energy metabolism.

#### 2.4.3. Inflammatory Response

Emerging evidence indicates that PBDEs may promote inflammation in mammalian tissues. For example, positive relationships between PBDEs and pro-inflammatory cytokines (IL-6 and TNF-α) in circulation were reported in pregnant and postpartum women [112]. Studies demonstrated that PBDEs modulate inflammatory pathways in the human placenta [113,114,115]. Proinflammatory cytokines have been associated with increased ROS production, a decrease in the production of testosterone in Leydig cells, and a decrease in sperm motility and concentration [116,117,118]. The role of PBDE-induced inflammation on male reproductive outcomes is not yet well understood.

For instance, inflammatory cell infiltration was observed in the epididymis interstitium of mice dosed with 30 mg/kg/BW of BDE-3 [52]. In another study, the suppurative inflammation of the epididymis in mice was observed after exposure to 30 mg/kg of BDE-47 for 6 weeks [54]. Additionally, the proteomic and metabolomic analysis of testis tissue showed that BDE-47 can trigger the apoptosis and inflammatory pathway [54]. However, this analysis was carried out for a merged list of proteins and metabolites affected by either dose used in the study: 1.5, 10, and 30 mg/kg/day daily for 6 weeks. Given that two higher doses in this study have very low relevance to real-life human exposure, we conducted Metascape enrichment analysis [119] for the list of proteins affected by the low dose, which resulted in no enriched biological categories relevant to inflammation.

An increase in the expression of inflammatory response genes was observed in the testes of adult rats after perinatal exposure to 0.2 mg/kg of BDE-47 [39] and in the testes of immature rats after prenatal exposure to 0.2 mg/kg of BDE-99 [44]. The biological categories enriched with upregulated genes included interferon, IL3 and IL5, and TNF-α signaling, allograft rejection, and natural killer cell-mediated cytotoxicity, among others. Additionally, exposed animals had significantly smaller testes, decreased sperm production, and an increased percentage of morphologically abnormal spermatozoa [39]. Similar changes in male reproductive function (decreased testis weight and an increased percent of morphologically abnormal spermatozoa) were reported in adult mice exposed to inflammatory challenge [120]. Taken together, these findings suggest that PBDE-induced changes in immune response genes in the testes may be causally linked with the disruption of male reproductive function.

#### 2.4.4. Disruption of Blood–Testis Barrier (BTB)

A growing body of literature provides clear evidence that early-life exposure to PBDEs at environmentally relevant doses may have long-lasting effects on mammalian tissue physiology [100,101,121,122], including male reproductive tissues [37,38,39,44,46]. Studies demonstrate that perinatal or neonatal exposures result in reduced testis weight and reduced sperm quality parameters [37,39,44,46,54].

One study demonstrated that exposure to BDE-209 may result in decreased activity of cortactin (CTTN) due to decreased CTTN expression and the increased Tyr phosphorylation of CTTN [46]. Upon activation, CTTN recruits Arp2/3 complex proteins to actin microfilaments, inducing actin branching from nucleation sites [123]. Thus, decreased CTTN activity in response to BDE-209 may disrupt ectoplasmic specialization [124,125], an actin-based junctional structure between the Sertoli cells and Sertoli cells (basal endoplasmic specialization) or Sertoli cells and germ cells (apical endoplasmic specialization) [126]. Ectoplasmic specialization is an essential mechanism supporting the blood–testis barrier (BTB), a structure that regulates the biochemical environment in the apical compartment of the seminiferous epithelium in which germ cells develop and protects germ cells from toxic compounds and autoimmune response [127]. Intact BTB is necessary to avoid the production of anti-sperm antibodies and autoimmune response leading to male infertility [128]. Damage of the BTB is associated with inflammation, germ cell loss, reduced sperm count, and ultimately subfertility or infertility [129]. From this point of view, it is interesting that in adult animals developmentally exposed to PBDE, inflammatory response genes were upregulated in the testes [39]. Similarly, the immune response pathways were enriched in the testes of immature rats prenatally exposed to 0.2 mg/kg body weight or higher doses of BDE-99 [44].

Thus, one hypothesis may explain the developmental male reproductive toxicity of PBDEs via their effects on the BTB formation and subsequent chronic inflammation caused by “leaky” BTB. One recent study report decreased the expression of tight junction proteins ZO-1 and β-catenin in mice testes following 8 weeks of adult exposure to 20 mg/kg/BW or higher doses of BDE-209 [56]. At higher doses (100 and 500 mg/kg/BW), the discontinued structure of the BTB junctions was observed in this study. Additionally, one study indirectly supports the BTB hypothesis, demonstrating that the gestational and lactational exposure of rats to 0.06 mg/kg/day of a mix of brominated flame retardants (technical PBDE mixtures DE-71, DE-79, and BDE-209, and HBCDD) significantly downregulated adherens junction proteins, E-cadherin, and β-catenin, and the gap junction protein connexin 43 (Cx43) in post-pubertal mammary glands [130]. These proteins play important roles in the BTB integrity, and changes similar to these observed in the mammary gland may cause BTB disruption in testes.

#### 2.4.5. Endocrine Disruption: Testosterone Signaling

PBDEs are well-recognized endocrine disruptive chemicals (EDCs). Specifically, a substantial body of literature connects PBDE exposures with reproductive hormone signaling.

PBDEs may have opposite effects on testosterone synthesis during peripubertal and adult exposures. For example, the exposure of rats to environmentally relevant doses of BDE-47 during the peripubertal window (PND21–35) resulted in Leydig cell hyperplasia, the increased expression of steroidogenesis enzymes in Leydig cells, and, ultimately, increased serum testosterone [38]. In another study, Leydig cells were obtained from prepubertal rats (PND49) and used to analyze the effects of BDE-47 in vitro. The testosterone production was three-fold higher at a 1 mM BDE-47 concentration compared with the control [131]. Similarly, in humans, developmental exposure to PBDE may increase testosterone secretion later in life. For example, in a birth cohort study, a 10-fold increase in maternal prenatal serum concentrations of BDE-153 was associated with an approximately 92.4% increase in testosterone in 12-year-old sons [132]. Not all studies support the increased production of testosterone following developmental exposure to PBDE. For example, in young prepubertal rats, the serum testosterone level significantly decreased following prenatal exposure to 0.2 mg/kg body weight BDE-99 and higher doses [44]. Similarly, 35-day-old mice had decreased blood testosterone and decreased numbers of Leydig cells following prenatal exposure to 0.2 mg/kg/BW and higher doses of BDE-99 [44].

In experiments where adult rats were exposed to low doses of BDE-47, the blood testosterone levels were significantly decreased in a dose-dependent manner [48,49]. Blood testosterone was also decreased in adult mice exposed to high doses of BDE-209 [55,133]. Decreased testosterone was also seen in mice exposed via mother’s milk to BDE-209, although the relevance of these findings is low due to the use of very high doses (500 and 700 mg/kg) [96]. Human data on testosterone levels in relation to adult exposures have only started to emerge. For example, in one study of 63 US men, positive associations of octa-BDE in house dust with serum testosterone and an inverse association of deca-BDE in house dust with testosterone were reported [134]. BDE-47 was also positively associated with testosterone levels in adult male sport fish consumers [21].

Many studies directly tested the ability of PBDEs to bind to androgen receptors (AR) using ex vivo, in vitro, and in silico experiments. Some of these studies analyzed the ability of individual PBDE congeners, their mixes, and their hydroxylated and methoxylated metabolites to interfere with androgen receptors using in vitro assays in which reporter cell lines carry a luciferase gene under the transcriptional control of response elements for activated AR (chemically activated luciferase gene expression (CALUX) assay) [50,135,136]. These studies did not identify any PBDE congener or metabolite with AR-agonistic activity [135,136]. However, they did report that many PBDEs have AR-antagonistic activity at physiologically relevant concentrations.

For example, one study of 19 PBDE congeners reported four congeners (BDE-19, BDE-100, BDE-47, and BDE-49) with IC_50_ values lower than IC_50_ for flutamide, a reference AR-antagonistic compound and nonsteroidal antiandrogenic drug [135]. Only three congeners did not show AR antagonistic activity in that study. Similarly, BDE-47, BDE-100, and industrial penta-BDE mixture DE-71 inhibited dihydrotestosterone (DHT)-induced AR activation by 50% at 5 μM concentrations in another study, while BDE-99, BDE-153, and BDE-154 did not show AR antagonism [50].

Another study reported the AR antagonistic activity of 12 out of 16 tested PBDE congeners and metabolites with IC_20_ in a range of 10 nM–1 μM [136]. The authors reported that the anti-androgenic activities of 4′-HO-BDE-17 and BDE-100 were about 5- and 10-fold lower than that of hydroxyflutamide, respectively. Specifically, the IC_20_ for 4′-HO-BDE-17 was determined to be as low as 0.086 μM. Similar experiments conducted by different research groups resulted in higher values for inhibitory concentrations of hydroxylated metabolites of BDE-47 [137]. In that study, the IC_50_ of 4′-HO-BDE-17 was 1.41 μM, and 6-HO-BDE-47 was identified as the most potent hydroxylated metabolite (IC_50_ = 0.34 μM). In a study of methoxylated PBDE metabolites, 6-MeO-BDE-47 was identified as a potent anti-androgen with IC_50_ = 41.8 μM [138].

The AR-antagonistic activity of PBDEs was also assessed ex vivo in a competitive binding assay with rat ventral prostate cytosol [50]. DE-71 and BDE-100 inhibited the AR binding of a radiolabeled synthetic androgenic steroid [^3^H]R1881, with an IC_50_ of approximately 5 μM [50]. An in silico study using an induced fitting dock test and binding affinity estimation showed that BDE-47, BDE-99, and their methoxylated and hydroxylated metabolites are tightly bound to the AR binding site in similar pattern to how testosterone binds to AR [139]. The methoxylated metabolites of BDE-47 and BDE-99 had higher binding energy than the parent compounds, and 6-MeO-BDE-99 had the highest binding energy among all tested metabolites, close to the binding energy of testosterone.

In the general population, PBDE concentrations in blood may reach the nM range [97,98,99], suggesting that in highly exposed individuals, PBDE and their metabolites may disrupt androgenic signaling by the direct antagonistic interaction with AR. Additionally, changes in testosterone signaling may result from its decreased production and/or altered hypothalamic–pituitary control.

#### 2.4.6. Endocrine Disruption: Estrogen Signaling

PBDE also demonstrates xenoestrogenic properties in a range of in vitro studies. We were not able to identify animal studies that provide unambiguous support of (anti)estrogenic properties of PBDEs. In the only identified human study, maternal serum BDE-145 at 35 weeks of pregnancy in a Dutch cohort was positively correlated with estradiol (E_2_) and free E_2_ levels in their sons at 3 months of age [140].

Several studies used estrogen receptor (ER)-CALUX assays in different cell lines to assess (anti)estrogenic properties of PBDE and their metabolites. The assessment of 17 PBDE congeners showed that 11 compounds have agonistic activity with concentrations leading to a 50% induction (EC_50_) ranging from 2.5 to 7.3 μM [141]. The potencies of these congeners were 250,000–390,000 times lower than the potency of the natural ligand, estradiol (E_2_). An ER-CALUX assay in the presence of E_2_ demonstrated that some hexa-BDE and one hepta-BDE have ER-antagonistic properties at IC_50_ in the 0.8–3.1 μM range. In an ERα-specific assay, 4′-OH-BDE-30 demonstrated estrogenic properties with an EC_50_ < 0.1 μM (four orders of magnitude lower than EC_50_ for E_2_) [141]. Another study tested the estrogenic activity of 12 PBDE congeners and their hydroxylated metabolites using ER-CALUX assay in MCF-7 cells [142]. Two hydroxylated metabolites, 4′-OH-BDE-17 and 3′-OH-BDE-7, but not other tested compounds, exhibited estrogenic activity at 1–10 μM concentrations. The estrogenic activity of OH-PBDE was also tested in vitro in two luciferase reporter gene systems in another study [143]. The EC_50_ of the most potent metabolite, 4′-OH-BDE-17, was determined as 4.7 μM, while the EC_50_ of E_2_ was 1.2 pM [143]. Another ER-CALUX study in MCF-7 cells 10 of the 22 OH-PBDEs exhibited luciferase induction at μM concentrations and demonstrated 10^5^- to 10^7^-fold smaller potency than E*_2_* [144]. Additionally, six HO-PBDEs showed ER-agonistic effects, four showed no effects, and twelve demonstrated ER-antagonistic effects. Interestingly, the ER agonists in these studies had four or fewer bromine atoms, while the ER antagonists had four or more bromine atoms [144].

ER-agonistic properties were also found for low-brominated PBDE, up to hexa-brominated BDE-155 in a study of 19 PBDE congeners and 6-OH-BDE-47 using ER-CALUX assay in T47D human breast cancer cells [135]. The EC_50_ values were >2 μM, six orders of magnitude lower than for E_2_. Tetra-congener (BDE-79), all tested hepta-congeners (BDE-181, BDE-183, BDE-185, and BDE-190), and 6OH-BDE-47 demonstrated ER-antagonistic properties. 6-OH-BDE-47 was the most potent anti-estrogen in this study, with IC_50_ = 0.5 μM, which is around 3000 times lower potency than that of a reference antiestrogenic drug ICI 182.780 [135]. The ER-agonistic properties of low-brominated PBDE and ER-antagonistic properties of higher-brominated PBDE were confirmed in another study carried out by the same research group [145]. In a study of eight PBDE congeners and their metabolites in ERα and ERβ CALUX assays in BG1Luc4E2 ovarian cancer cells, three (BDE-28, BDE-47, and BDE-100) showed ERα-agonistic properties, BDE-100 also showed ERβ-antagonistic properties, and two other congeners (BDE-99 and BDE-153) showed antagonistic properties for both ER receptors in the μM range [136]. Among the metabolites, 4′-HO-BDE-17 showed the most potent estrogenic activity (EC_50_ = 0.2 μM for both ERα and ERβ assays), and 4′-HO-BDE-49 showed the most potent anti-estrogenic properties (EC50 = 2.3 and 3.6 μM for ERα and ERβ assays, respectively) [136]. Among the BDE-47 hydroxylated analogs, 4′-HO-BDE17 induced a significant estrogenic response, while other compounds showed anti-estrogenic potency (4′-HO- BDE-17, 6-HO-BDE-47, 2′-HO-BDE-28, BDE-47) or no (anti)estrogenic activity (4′-HO-BDE-49) in the micromolar range [137].

The direct binding of 22 OH-PBDEs with the human ERα ligand-binding domain was measured using a surface plasmon resonance technique using E_2_ as the positive control [144]. The K_D_ (equilibrium dissociation constant) of E_2_ was 0.35 nM in this study. Seven out of the twenty-two OH-PBDEs showed a direct binding reaction at the ERα ligand-binding domain with K_D_ values in the range of 0.15–7.90 μM. The relative binding potency of the most potent compound, 6-OH-BDE-47, was 0.24% of E_2_. Six OH-PBDEs, metabolites of DE-71, were tested in a competitive binding assay with recombinant ERα and tritiated E_2_ (^3^H-E_2._) [143]. All of the OH-PBDE displaced ^3^H-E_2_ from ERα, but their binding affinities in relation to E2 ranged from 0.001% to 0.03%. 4′-OH-BDE-17 and 4′-OH-BDE-49 were the most potent metabolites, with IC_50_ in the micromolar range.

Xenobiotics may induce (anti)estrogenic singling via both canonical nuclear estrogen receptors (ERs) as well as via nongenomic G protein-coupled estrogen receptor (GPER) pathways [146]. The ability of PBDE to induce nongenomic estrogenic signaling was assessed in a recent study where the binding affinities of 12 PBDE congeners and their 18 hydroxylates metabolites to GPER were determined in a competitive binding assay in vitro. Eleven hydroxylated PBDEs, but none of the PBDEs, bound to GPER directly with EC_50_ ranging from 1.3 to 20 μM and relative binding affinities ranging from 1.3% to 20.0% compared to E_2_ [147].

According to another hypothesis, PBDEs may exert estrogenic properties via a non-ER-dependent mechanism via the inhibition of estradiol sulfotransferase enzymes [148]. The decreased sulfation of E_2_ may result in an increased bioavailability of natural estrogens and increased estrogenic signaling. The ability of a recombinant human sulfotransferase 1E1 (SULT1E1) to produce E_2_ sulfate from tritiated E_2_ in the presence of PBDE congeners and their hydroxylated metabolites were tested ex vivo [135,145,148]. Most PBDE congeners inhibited SULT1E1 at μM concentrations, while OH-PBDEs had IC_50_ in a range of 0.2–1.4 μM.

The studies discussed in this section demonstrate that (1) low-brominated PBDEs and their metabolites mostly show estrogenic activity, while highly brominated compounds demonstrate anti-estrogenic properties; (2) hydroxylation mostly increases the estrogenic or anti-estrogenic properties of PBDE; (3); PBDEs and their metabolites may interact with estrogenic signaling via a different mechanism at the μM range; and (4) PBDEs and their hydroxylated metabolites have activities towards ERs 10^4^ to 10^7^-fold less potent than E_2_. These findings, taken together with the lack of populational studies indicating changes in circulating estrogens in relation to PBDEs, suggest that the disruption of estrogenic signaling is an unlikely mechanism of the male reproductive toxicity of PBDEs.

#### 2.4.7. Endocrine Disruption: Luteinizing Hormone (LH) Signaling

Human studies indicate that early-life exposure to PBDEs results in increased LH secretion later in life. For example, in a birth cohort study, a 10-fold increase in maternal prenatal serum concentrations of BDE-100 and BDE-153 were associated with approximately 75% and 97% increases in LH in 12-year-old boys, respectively [132]. Similarly, the amount of PBDEs in maternal milk positively correlated (*p* < 0.033) with serum LH levels in three-month-old infants in a prospective Danish–Finnish study [149]. Additionally, serum LH levels correlated positively with individual congeners BDE-47, BDE-100, and BDE-154. A rat study demonstrated the opposite relationship between developmental PBDEs and LH production, although this study used a different exposure window (prepubertal) [38].

Studies of LH association with adult exposures produce controversial results. For example, in a study of 24 men recruited through a US infertility clinic, the concentrations of BDE-47, BDE-99, and BDE-100 in house dust were inversely associated with serum LH [150]. However, in an extended study of 63 participants by the same group, these associations were not significant [134]. Additionally, the study reports positive associations between the dust concentrations of octa-BDEs and serum LH [134]. In a study of 27 adult US men, LH serum concentrations were positively associated with BDE-47 and BDE-99; however, the positive relationships were almost absent following the removal of a single influential data point [151]. Similarly, no association between PBDEs and LH was observed in 77 men working in e-recycling facilities or other recycling facilities (low-exposure group) [152]. In another e-recycling facility study of 76 men in southern China, a positive correlation was found between semen or serum levels of BDE-153 and serum LH [153].

Several rodent studies analyzed LH response to PBDE exposures. For example, blood LH was also increased in adult male rats exposed to 3 and 30 mg/kg DE-71 [50] and in 35-day-old mice exposed perinatally to 20 mg/kg/BW of BDE-99 [44].

#### 2.4.8. Endocrine Disruption: Follicle-Stimulating Hormone (FSH) Signaling

There are only a few studies that report FSH levels in response to PBDE exposure in population studies, and their findings are conflicting. In a birth cohort study, a 10-fold increase in maternal prenatal serum concentrations of BDE-153 was associated with approximately 22% increases in FSH in 12-year-old boys, respectively [132]. Other studies analyzed FSH in response to PBDE in adult subjects. In a study of 24 men recruited through a US infertility clinic, the concentrations of PBDE in house dust were inversely associated with serum FSH [150]. A reanalysis in an extended study of 63 participants demonstrated a significant inverse association between the dust concentrations of penta-BDEs and serum FSH [134]. In a southern China study of 54 highly exposed and 58 moderately exposed participants, negative associations between serum FSH and serum concentrations of several PBDE congeners were found in female, but not male, participants [154]. Another study of 27 USA men reported the opposite relationship between PBDE and FSH: BDE-47, BDE-100, and BDE-153 were significantly positively associated with FSH among older men (≥40 years old), but not younger men (<40 years old) [151]. No association between PBDEs and FSH was observed in 77 men working in e-recycling facilities or other recycling facilities (low-exposure group) in southern China [152]. One recent study report increases of blood FSH in 35-day-old mice following prenatal exposure to 20 mg/kg/BW of BDE-99 [44].

#### 2.4.9. Endocrine Disruption: Inhibin-B and Sex Hormone-Binding Globulin (SHBG)

There are only a few studies that report inhibin B and/or SHBG levels in response to PBDE exposure in population studies. The amount of PBDEs in maternal milk did not correlate with the serum levels of inhibin B and SHBG in three-month-old infants in a prospective Danish–Finnish study [149]. However, in another study from the Netherlands, maternal serum BDE-145 at 35 weeks of pregnancy positively correlated with inhibin B levels, but not SHBG, in their sons at 3 months of age [140].

In a study of 24 men recruited through a US infertility clinic, concentrations of BDE-47, BDE-99, and BDE-100 in house dust were positively associated with inhibin B and SHBG [150]. SHBG was also positively associated with penta-BDE in an extended study of 63 participants, while the association of inhibin B with PBDEs became non-significant [134]. In a study of 27 USA men, serum PBDEs were inversely associated with inhibin B [151]. No associations between PBDEs and SHBG were found in this study.

#### 2.4.10. Thyroid Hormone Signaling

Thyroid hormones play an important role in male reproductive health. Hypothyroidism is associated with impaired concentrations of reproductive hormones and abnormalities in sperm morphology, while thyrotoxicosis induces abnormalities in sperm motility [155,156]. Thyroid hormones (TH) inhibit Sertoli cell proliferation [157] and promote their differentiation [155]. They also suppress the tight junctions between Sertoli cells and spermatogonia [158]. Additionally, thyroxine (T_4_) has a direct effect on sperm motility [159]. Overall thyroid dysfunction may result in reduced fertility and infertility [155]. Altogether, this evidence suggests that thyroid disruption by PBDE may have multifaceted effects on male reproduction.

Due to the structural similarity of PBDEs and thyroid hormones, the ability of PBDEs to affect thyroid signaling has attracted a lot of attention from the research community. A substantial body of evidence exists to date demonstrating thyroid disruption by PBDEs in the general population and in laboratory animals at environmentally relevant doses. For example, the pre- or perinatal exposure to low doses of BDE-47 resulted in significant decreases in T3 and T4 in rats [37,160] and sheep [161]. Additionally, T3 was negatively correlated with sperm counts [37].

Human population studies produce controversial results on the associations between circulating THs and PBDE exposure. For example, a significant correlation between serum PBDE and thyroid dysfunction was documented in a recent population study in China [162]. A recent meta-analysis of 16 population studies concluded that serum THs were negatively associated with serum PBDEs when the median levels of PBDEs were <30 ng/g lipid; there was no correlation between THs and PBDEs at median levels between 30 ng/g and 100 ng/g lipids, and the associations were mostly positive if the median levels of PBDEs were >100 ng/g lipids [163]. These potentially U-shaped dose–response relations within the range of human exposure in the general population make the analysis of the role of thyroid disruption by PBDEs on male reproductive health particularly complex.

PBDE can likely affect thyroid signaling via several molecular mechanisms. The most well-documented mechanism consists of TH displacement by PBDEs and their metabolites from TH transport proteins. For example, the binding constants of 14 OH-PBDEs with transthyretin (TTR) and thyroxine-binding globulin (TBG) assessed by competitive fluorescence displacement assay were in a range of 69–140 nM for TTR and 22 nM–6.5 μM for TBG, with tetrabrominated compounds having the highest binding affinity [164]. The dissociation constant (K_d_) for TTR and 11 OH-PBDEs were in the nM range in another study using different methodological approaches to analyze the binding affinities of PBDE with TH transport proteins. [165]. In this study, the K_d_ of 10 and 5 OH-PBDE were lower than the K_d_ of T3 and T4, respectively. For TBG, the K_d_ of 7 OH-PBDE was also in the nM range and comparable with the K_d_ of THs. Other studies support these findings [135,166,167]. Additionally, PBDEs and their metabolites may affect other aspects of thyroid signaling, including the hypothalamic–pituitary control of thyroid signaling, the conversion of T_4_ to T_3_ by deiodinases in tissues, and the interaction with multiple nuclear receptor isoforms [168,169,170].

The well-documented ability of PBDE and their metabolites to disrupt thyroid signaling in the general population together with experimental toxicological evidence and mechanistic studies demonstrate that thyroid disruption is a plausible mechanism of the male reproductive toxicity of PBDEs. The detailed discussion of the mechanisms of thyroid disruption by PBDEs is outside the scope of the current study.

#### 2.4.11. Insulin-like Growth Factor (IGF) and Mechanistic Target of Rapamycin (mTOR)

IGF-1 is another important metabolic hormone that is poorly studied both in relation to men’s reproductive health and as a target of endocrine disruption by PBDEs. Understanding of the role of the growth hormone (GH)/IGF system in human reproductive physiology started to emerge only recently [171]. IGFs participate in sexual differentiation during fetal development and promote puberty onset [172]. Lower IGF-I levels are associated with impaired sperm parameters [173].

Two studies done in rats and mice demonstrated that perinatal exposures to low, environmentally relevant doses of BDE-47 result in a long-lasting increase in plasma IGF-1 [100,174]. Interestingly, in a mouse study, a two-fold increase in the circulating IGF-1 was observed in males on PND140, following perinatal exposure to 0.2 mg/kg of BDE-47 [100], suggesting permanent changes in IGF-1 signaling following early life programming.

Human data are scarce. For example, a positive correlation was observed between umbilical cord blood PBDEs levels and placental expression of insulin-like growth factor-binding protein 3 (IGFBP-3), the major IGF-1 transport protein in the blood [175]. In addition, a significant positive correlation was observed between BDE-154, BDE-209, and IGF-1 mRNA levels in the placenta. A cord blood log of IGF-1 levels was also positively correlated with the log of BDE-196 level in breast milk and negatively with the log of BDE-85 in breast milk [176].

The binding of IGF-1 to its respective receptor IGF-1R triggers a PI3K/Akt cascade, resulting in the activation of the mechanistic target of rapamycin (mTOR) complex one (mTORC1) and two (mTORC2). Therefore, the ability of PBDEs to induce mTOR shown in studies discussed below likely provides additional evidence of IGF-1 disruption.

The mTOR-centered pathway is a metabolic master-switch, which, at starvation, suppresses biosynthetic programs and increases the recycling of proteins and organelles to provide an internal resource of metabolites [177]. Conversely, the stimulation of the pathway by nutrients and growth factors causes the activation of biosynthesis and the suppression of autophagy [178]. According to the human protein atlas, the highest levels of mTOR RNA expression are found in the testes and the highest levels of mTOR protein expression are found in the prostate in comparison with all other human tissues [179]. Patients using mTORC1 inhibitors for immunosuppression therapy experience spermatogenesis disruption [180,181,182,183].

In the testes, the mTOR pathway plays multiple roles. One important tissue-specific role of the pathway consists in the regulation of the BTB integrity. Recent studies have shown that BTB integrity is determined by the balance between the activities of the two mTOR complexes, with mTORC1 promoting disassembly of the BTB and mTORC2 promoting its integrity [184,185]. The inhibition of mTORC1 by rapamycin resulted in BTB strengthening [186]. It was demonstrated further that ribosomal protein S6 (rpS6), the downstream phosphorylation target of mTORC1, is essential for F-actin network restructuring in Sertoli cells [186]. The same research group demonstrated that the knockdown of Rictor (a key component of mTORC2) in Sertoli cells in vitro was associated with a loss of barrier function, changes in F-actin organization, and loss of interaction between actin and adhesion proteins [187].

We were not able to identify studies in which the PBDE disruption of mTOR signaling was analyzed in male reproductive organs. Studies of other organs and cell types demonstrate mTOR modulation by PBDEs. For example, the developmental exposure to 0.2 mg/kg of BDE-47 activated both mTORC1 and mTORC2 in mouse livers, as measured by the phosphorylation of phospho-Akt (ser473) and phospho-p70S6 kinase (Thr389), respectively [100]. In the same study, the activity of both mTOR complexes was significantly induced following the exposure of the human hepatoma cell line (HepG2) to 1 μM BDE-47. A recent study from another research group reported the activation of PI3K/Akt signaling, which is an upstream activator of mTOR, by BDE-47 and BDE-85 in pancreatic beta-cells [188]. mTOR was also activated in human-derived hepatic cells (HepaRG) by 25 μM of BDE-47 or BDE-99 [189] and in the human fetal hepatocyte line (LO2) by 50 μM of BDE-209 [91]. Conversely, the treatment of human preadipocytes with 1 or 3 μM of BDE-28 resulted in the decreased phosphorylation (activity) of several key players in the mTOR pathway [105].

Interestingly, a reanalysis of published transcriptomic changes in response to PBDE exposure identified changes in ribosomal genes as a molecular signature of exposure [101]. Given that ribosomal genes are controlled by the mTOR pathway [190,191], their coordinated changes may be used as a surrogate measure of mTOR activity. It was shown that 0.2 mg/kg body weight of BDE-47 increased the expression of ribosomal genes in different species (mice, rats) and different tissues (liver, brain frontal lobes). However, the effects of higher doses of BDE-47 or DE-71 were the opposite and corresponded to mTOR suppression [101].

Overall, the role of IGF-1 disruption by PBDE in male reproductive health remains poorly understood. However, the importance of the IGF-1-activated mTOR pathway for testes physiology as well as direct evidence of mTOR modulation by PBDEs suggests the potential involvement of IGF-1/mTOR signaling in PBDE-induced male reproductive toxicity.

#### 2.4.12. Disruption of Steroidogenesis

Steroidogenesis is the process by which cholesterol is converted into biologically active steroid hormones, including sex steroids [192,193]. The steroidogenic process is regulated mainly by the transcription of genes that encode steroidogenic enzymes and co-factors [194]. For instance, testicular steroidogenic acute regulatory (StAR) protein plays an essential role in the transport of cholesterol to the inner mitochondrial membrane [195], the place where CYP11A1 converts cholesterol to pregnenolone, further used in the endoplasmic reticulum for the production of the final steroid product [196]. Many studies demonstrated that PBDEs and their metabolites affect various steroidogenic enzymes.

The effects of PBDEs on steroidogenic activity were first demonstrated in human adrenocortical carcinoma cells (H295R). An analysis of the expression of ten steroidogenic genes in H295R cells following exposures to DE-71, DE-79, and 20 different PBDE metabolites at 0.025, 0.05, and 0.5 μM concentrations demonstrated the ability of the majority of the tested compounds to change the expression of steroidogenic enzymes with four metabolites (6-MeO-BDE-47, 6-OH-BDE-90, 2′-OH-BDE-68, and 6-OH-BDE-47) producing significant changes at the lowest tested dose [197]. Similarly, most of the PBDEs and their metabolites activated steroidogenic enzymes in another study with the H295R cell line following exposure to hydroxylated and methoxylated PBDEs [198]. Specifically, CYP19 (aromatase) was activated by many metabolites at a 10 μM concentration.

Capitalizing on these studies, the researchers in [199] used a primary culture of rat Leydig cells to develop a mechanistic hypothesis connecting PBDE exposure with altered testosterone production. They hypothesized that PBDE may interfere with the cAMP pathway, which is known to regulate the rate-limiting enzymes of androgen biosynthesis, StAR, and CYP11A1 [200]. To test this hypothesis, Leydig cells were exposed to DE-71 at doses covering approximately the range between 1 and 30 nM. DE-71 stimulated testosterone secretion at 10 nM and higher concentrations. Increased steroidogenesis was associated with the increased production of cAMP, increased expression of StAR, and increased enzymatic activity of CYP11A1. Similarly, the significant upregulation of steroidogenic genes (SCARB1, StAR, and HSD11B1) was observed in Leydig cells cultured with 100 nM BDE-47 in another study [38]. The co-administration of adenylyl cyclase (AC) inhibitor prevented DE-71-stimulated testosterone secretion, suggesting that PBDE regulates (stimulates) steroidogenesis at some mechanism upstream of the cAMP pathway [199]. Interestingly, the increased production of cAMP positively regulated mTORC1 via the PKA-dependent pathway [201,202], suggesting that the mTORC1 activation discussed in this review and increased steroidogeneses may result from the same upstream events triggered by PBDEs.

The expression of several genes involved in testosterone synthesis (Star, Cyp11a1, Hsd3b1, Cyp17a1, and Hsd17b3) was decreased to varying degrees in prepubertal testes following prenatal exposure to 0.2 mg/kg body weight BDE-99 and higher doses in rat [44] and mouse studies [44]. These changes matched the reduced testosterone levels in circulation. Two studies using high doses of BDE-209 also showed the suppression of steroidogenic enzymes in mouse testes [96,133]. These in vivo results seemingly contradict in vitro data discussed in the previous paragraph. This contradiction may originate from the opposing effects of different doses of PBDE. For example, in a study with another type of steroidogenic cells, bovine luteal cells, BDE-99 stimulated the production of progesterone at 0.1–0.3 μM concentrations, but suppressed it at 1–3 μM concentrations [203]. Given that the threshold between the activating and suppressive concentrations of PBDE could be congener-specific, current knowledge does not enable the prediction of the effects of different PBDE mixes and concentrations on steroidogenesis. However, existing knowledge allows the suggestion that, overall, low doses of PBDE stimulate steroidogenesis and higher doses suppress it.

#### 2.4.13. Mitochondria Disruption and Cell Apoptosis

Mitochondria play a critical role during the process of spermatogenesis [204]. Apart from their role in energy (ATP) production crucial for secretory activity [205], sperm motility [206], and other physiological aspects of reproductive tissues, mitochondria are the major site for steroid hormone production [207] and they play important roles in cell signaling, cell proliferation, and death [208,209]. Specifically, anti-apoptotic and pro-apoptotic proteins in the mitochondria recruit and activate the caspase cascade to regulate testicular apoptosis during the process of spermatogenesis [210].

Several authors hypothesized that the male reproductive effects of PBDE may be mediated via the disruption of mitochondria, which may lead to apoptosis [45,211]. Most of the studies providing evidence in support of this hypothesis report mitochondria-related outcomes at doses significantly exceeding environmental exposure.

For example, the analysis of mitochondrial outcomes in immortalized mouse spermatocytes exposed to 0.1, 1, 10, and 100 μM of BDE-47 for 48 h showed an altered mitochondrial ultrastructure at the highest dose, the decreased expression of mitochondrial proteins Atp5b and Uqcrc1, and decreased Bcl-2, an anti-apoptotic factor of the mitochondrial apoptotic pathway at the two highest doses and altered MMP and decreased ATP production at the three highest doses [211]. Overall, no mitochondrial disruption was found at the 0.1 μM concentration of BDE-47, suggesting that the doses inducing mitochondrial toxicity exceed the internal doses in the general population by around three orders of magnitude. In cultured rat Leydig cells, 10 μM BDE-99 induced the mitochondrial apoptotic pathway [44]. Similarly, the exposure of rat pheochromocytoma PC12 cells to 10 μM or higher concentrations of BDE-47 resulted in the inhibition of mitochondrial fusion (Mfn1 and Mfn2) and fission proteins (Fis1 and phosphorylated Drp1), decreased production of ATP, dissipation of mitochondrial membrane potential (MMP), mitochondrial fragmentation, and activation of apoptosis, while no mitochondrial disruption was induced by 1 μM of BDE-47 [212]. In the study of HepG2 cells exposed to a range of BDE-47 or BDE-154 doses, BDE-47 induced markers of mitochondria disruption and apoptosis at 1 μM and higher doses, while most of the effects of BDE-154 were observed at higher doses (25 μM) [213]. Both flame retardants failed to produce effects at 0.5 μM doses. The low sensitivity of mitochondria to PBDE exposure was also demonstrated in an in vivo study in which sperm MMP was reduced in mice exposed to 500 and 1500 mg/kg body weight BDE-209, but not at smaller doses (10 and 100 mg/kg body weight) [45].

## 3. Materials and Methods

Our review consists of two major parts. The first follows a systematic approach to identify male reproductive outcomes sensitive to PBDE in humans and animal studies. The second part is a narrative review of the molecular mechanisms involved in male reproductive toxicity (Figure 2).

### 3.1. Identification of Male Reproductive Outcomes Sensitive to PBDE

A search was conducted in PubMed on the 7th of November 2022 using the keyword “PBDE” in combination with one of the following: sperm, semen, prostate, testes, epididymis, spermatozoa, and male fertility. Studies non-relevant for our review were excluded following an analysis of the papers’ abstracts. Only original studies that investigated the male reproductive effects of PBDEs in humans and mammalian model organisms were included in the review.

In our previous research, we compared human and laboratory animal exposures using PBDE burdens accumulated in adipose tissue and concluded that doses in a range of a few milligrams per kilogram of body weight (BW) and lower may be considered relevant to human exposure in the general population [214]. Thus, in the current review, exposure doses <50 mg/kg/BW for animal studies were set as a conservative quality threshold to narrow down our analysis to only experiments that provide information relevant to human health. Around 30% of the systematically selected articles were rejected based on these criteria. However, these studies were retained for the narrative review of the mechanisms affected by PBDE. Human studies in which average sperm parameters deviated significantly from the standard population values in subjects recruited from the general population were filtered out, as this deviation may indicate (1) selection bias and/or (2) flaws in the methodology of sperm analysis. One human study was excluded from analysis based on these quality criteria. Studies that passed our search and filtering criteria (22 animal studies and nine human studies) were further reviewed to identify doses of PBDE that produced changes in male reproductive outcomes (Table 1 and Table 2).

### 3.2. Narrative Approach

In the narrative approach, we discuss mechanisms involved in the male reproductive effects of PBDEs. For this part, we first analyzed papers selected in the course of the systematic step of this study. Additional papers were analyzed if needed to add knowledge on the molecular mechanisms affected by PBDE from in vitro, ex vivo, and in silico studies and studies not focusing on male reproduction.

## 4. Conclusions

PBDEs were not created by natural selection or human intelligence to target specific molecular pathways in human cells. Thus, it is highly unlikely that PBDEs by chance have very high specificity to a single molecular target and do not interact with others. In other words, it is more likely that PBDEs affect multiple molecular pathways. This situation is not unique in environmental toxicology. For example, thousands of published papers on polychlorinated biphenyls and bisphenol A did not result in the identification of one critical pathway of reproductive or other toxicity of these compounds, but rather demonstrated that almost every tested molecular mechanism may be affected by these chemicals.

The non-specific nature of PBDEs creates significant uncertainty in relation to the understanding of the molecular mechanism of the reproductive toxicity of PBDEs. Based on the review of a significant body of literature, we identified a broad range of highly overlapping mechanistic hypotheses supported by evidence. These hypotheses include the induction of oxidative stress, metabolic disruption, inflammatory response, BTB integrity, endocrine disruption of reproductive hormones (testosterone, estrogen, LH, FSH, SHBG, and inhibin), thyroid signaling, IGF-1/mTOR signaling, disruption of steroidogenesis, and mitochondria disruption.

For a better understanding of the role of each mechanism, we suggest using exposure doses as a criterion that may help prioritize mechanisms by their relevance. Indeed, the ability of a chemical to affect some molecular mechanism at a dose corresponding to exposures of the general population is an indicator that the corresponding mechanisms may be causatively involved in the development of observed adverse reproductive outcomes associated with exposure. Conversely, if some molecular mechanism may only be disrupted by doses much higher than the exposures of the general population, this mechanism should be considered as having low relevance to the observed reproductive toxicity in the population, even if it is well recognized as a mechanism linked causatively with reproductive health.

Following this logic and the evidence discussed in this paper, we conclude that oxidative stress, the endocrine disruption of estrogenic signaling, and mitochondria disruption are likely mechanisms not affected by PBDE at doses relevant to environmental exposures in humans. The disruption of reproductive hormones LH, FSH, SHBG, and inhibin by PBDEs is still poorly understood. The existing data evidence does not yet build enough ground to evaluate the relevance of these mechanisms for the male reproductive toxicity of PBDEs in the general population.

Finally, published research provides evidence that another group of mechanisms may be affected at exposure levels equivalent to the exposure of humans in the general population (Figure 3). These mechanisms include metabolic disruption, inflammatory response, BTB integrity, the disruption of testosterone signaling, thyroid signaling and IGF-1/mTOR signaling, and the disruption of steroidogenesis. Interestingly, except for PBDEs binding to AR, other mechanisms in this group are highly overlapping. The dominating theme of these overlapping mechanisms is “metabolism”. Indeed, thyroid and IGF-1/mTOR signaling represent major master-regulators of cellular metabolism, and changes in these signaling pathways affect the metabolism of lipids and carbohydrates. Steroidogenesis itself is one of the metabolic pathways dependent on the availability of cholesterol and regulated upstream by some mechanisms common with mTOR upstream regulation. Additionally, mTOR regulates the immune response and is involved in the regulation of the BTB. Breaches in the BTB induce an autoimmune inflammatory reaction in the testes.

To conclude, our analysis of a broad range of mechanistic hypotheses suggests that the direct binding of PBDEs and their metabolites to AR receptors and the disruption of a complex network of metabolic signaling pathways are two major mechanisms responsible for the observed PBDE-induced decline in male reproductive health parameters. The triggering events for the disruption of the metabolic network likely include, but are not limited to, the binding of PBDEs to thyroid-transporting proteins. In addition to reproductive toxicity, the disruption of this network by PBDEs may contribute to metabolic disease, neurotoxicity, and other organ toxicities.

## Figures and Tables

**Figure 1 ijms-23-14229-f001:**
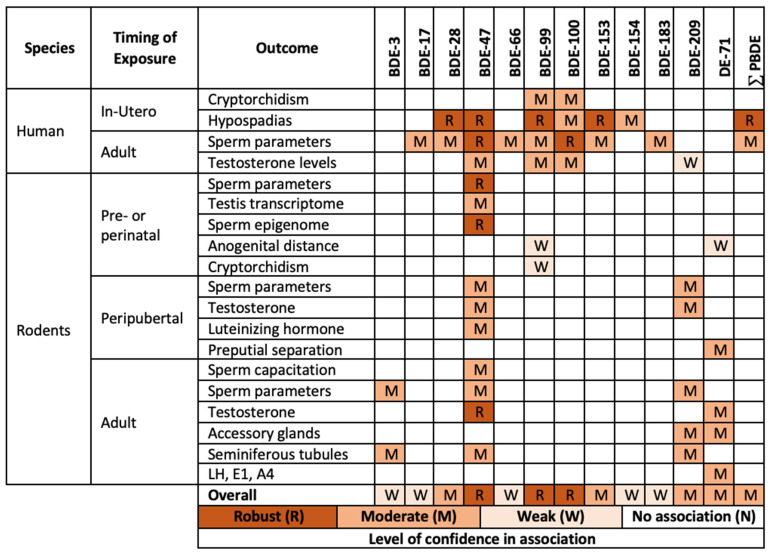
Summary of epidemiologic and experimental evidence of male reproductive effects associated with PBDEs. Findings may be biased due to different coverage of PBDE congeners by published research.

**Figure 2 ijms-23-14229-f002:**
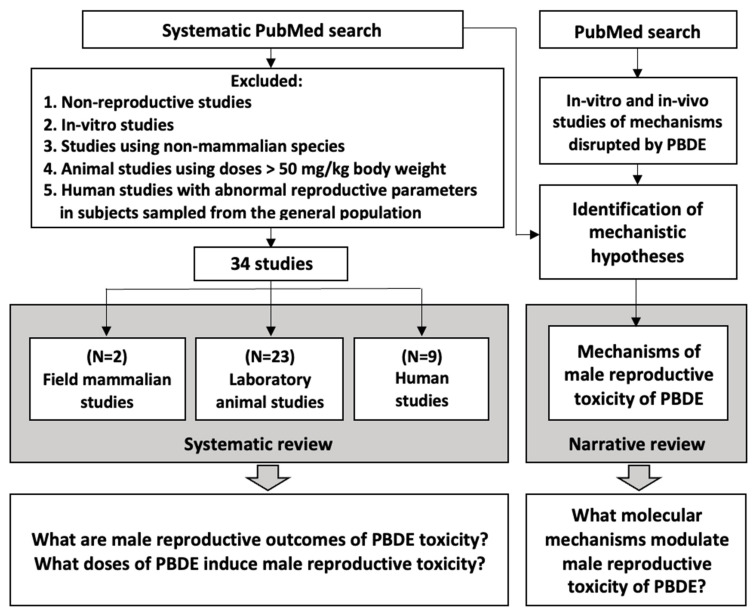
Literature flow diagram for disruption of male reproduction by PBDEs.

**Figure 3 ijms-23-14229-f003:**
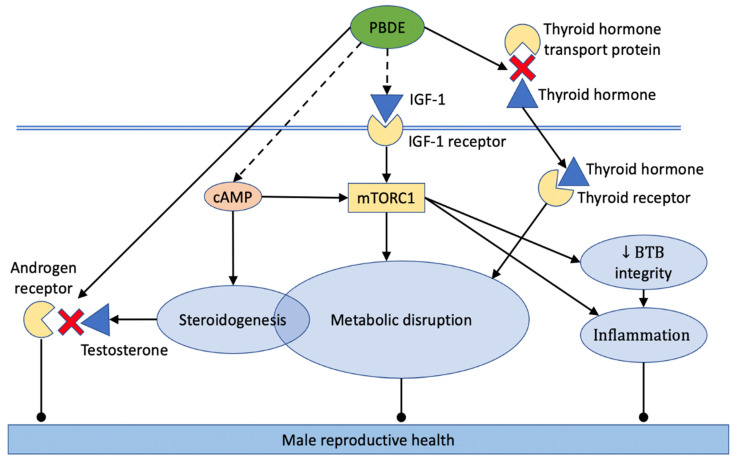
Summary of mechanisms involved in male reproductive toxicity of PBDE. (1) Many PBDEs and their metabolites may disrupt testosterone signaling via antagonistic interaction with the androgen receptor. (2) PBDEs increase the production of cAMP via an unknown mechanism. cAMP is a central regulator of steroidogenesis and altered cAMP production may result in altered availability of testosterone. cAMP is also involved in mTORC1 activation. (3) PBDEs increase circulating IGF-1, which activates mTORC1 signaling. mTORC1 positively regulates anabolic metabolic pathways and immune response. mTORC1 activity also promotes permeability of the blood–testes barrier. Increased permeability of the blood–testes barrier may result in autoimmune response and testicular inflammation. (4) PBDE and their metabolites compete with thyroid hormone for thyroid hormone transporting proteins in the blood. Replacement of thyroid hormone from transporting proteins alters the bioavailability of hormones and affects thyroid signaling.

**Table 1 ijms-23-14229-t001:** Summary of human studies assessing male reproductive toxicity of PBDEs.

PBDE Levels			Sample Size	Design and Timing of Exposure	Affected Outcomes	Timing of Outcomes	Study Population	References
Congeners	Concentration	Media						
10 BDEs, including BDE-17, 28, 153	Individual: range of medians (ng/g): 0.00–0.12	Serum	468 men (LIFE study)	Cross-sectional part of prospective study, mean age 31.8 years, 2005–2009	Negative association with sperm motility and morphology	Mean age 31.8 years	Texas and Michigan states	[25]
29 BDEs, including BDE-47, 99, 100, 153	Individual: BDE-47 median (ng/g lipid): 0.72	Serum	10 men	Cross-sectional, age 18–22 years, 2003	Inverse correlation with sperm concentration and testis size	Age 18–22 years	Kawasaki, Japan	[26]
7 BDEs, including BDE-47, 153	Individual: medians (ng/g lipid): BDE-47-Ukraine-0.2, Poland-0.6, Greenland-2.0; BDE-153-Ukraine-0.3, Poland-0.5, Greenland-2.7	Serum	299 men	Cross-sectional, IQR age: Ukraine 20.7–38.2; Poland 25.3–36.9; Greenland 21.2–43.6 years 2002–2004	No effect on sperm quality and serum reproductive hormones	IQR age: Ukraine 20.7–38.2; Poland 25.3–36.9; Greenland 21.2–43.6 years	Ukraine (Kharkiv), Poland (Warsaw), and Greenland	[27]
BDE-47, 99, 100	Total: median (ng/g lipid): 29.1 Individual BDE-47: median: 9.4	Hair	153 men	Cross-sectional, age 18–41 years, 2009–2012	Negative association with sperm motility	Age 18–41 years	Montreal, Canada	[28]
BDE-28, 47, 99, 100, 153, 154, 183, 209	Total: median (ng/g): 53.0. Individual: medians: BDE-99-7.9; BDE-100-7.4; BDE-154-4.0	Hair	137 mothers/boys vs. 158 controls	Case-control, age 18–48 years, 2011–2014	Higher risk of cryptorchidism	Antenatal, age 3–12 months	Montreal, Canada	[29]
BDE-28, 47, 153	Individual: range of medians (pg/g ww): 3.6–6.1	Semen	32 men	Cross-sectional, age 20–50 years, 2015–2016	Negative correlation with sperm concentration and count	Age 20–50 years	Qingyuan, China	[30]
BDE-28, 47, 99, 153, 154	Individual: range of 5 medians (ng/g): 4–10.8	Hair	152 mothers/boys vs. 64 controls	Case-control, mothers’ IQR age 29–36 years, 2011–2014	Higher level in cases of hypospadias	Antenatal, IQR age 5–12 months	Toronto, Canada	[31]
BDE-28, 47, 99, 100, 153, 154, 183, 209	Total: median (ng/g): 51.4	Hair	89 mothers/boys vs. 54 controls	Case-control, mothers’ IQR age 29–36 years, 2011–2013	Higher risk of hypospadias	Antenatal, IQR age 5–12 months	Toronto, Canada	[32]
BDE-28, 47, 99, 100, 153	Individual: range of 5 medians (ng/g lipid): 1.0–19.1	Serum	20 mothers/boys vs. 28 controls	Nested case-control, maternal mid-pregnancy, 2003	No effect on hypospadias	Antenatal	Southern California	[33]

**Table 2 ijms-23-14229-t002:** Summary of studies assessing male reproductive toxicity of PBDE in laboratory rodents.

PBDE	Lowest Toxic Dose, mg/kg Body Weight	Species	Route of Exposure	Duration of Exposure	Outcome Altered at Lowest Dose	Outcome Assessed	References
**Developmental Studies**							
BDE-47	0.2	Rat	Pipette feeding	GD8–PND21	DNA methylation of sperm	PND65 and PND120	[34]
BDE-47	0.2	Rat	Pipette feeding	GD8–PND21	Sperm small noncoding RNA	PND65 and PND120	[35]
BDE-47	0.2	Rat	Pipette feeding	GD8–PND21	DNA methylation of sperm	PND65 and PND120	[36]
BDE-47	0.1	Rat	Oral gavage	10 days before mating–PND21	Testis weight	PND88	[37]
BDE-47	0.4	Rat	Oral gavage	PND21–35	Leydig cell number, serum LH and testosterone levels	PND35	[38]
BDE-47	0.2	Rat	Pipette feeding	GD8–PND21	Sperm parameters, testes weight, daily sperm production, and testis transcriptome.	PND120	[39]
DE-71	30	Rat	Oral gavage	PND23–53	Preputial separation	PND53	[40]
DE-71	18	Rat	Oral gavage	GD6–PND18	Testis weight	PND31	[41]
BDE-99	0.06	Rat	Oral gavage	GD6	Daily sperm production, sperm and spermatid counts	PND140	[42]
DE-71	40	Rat	Oral gavage	GD7–PND16	Anogenital distance	PND1	[43]
BDE-99	0.2	Mouse	Oral gavage	GD1–GD21	Anogenital distance, testosterone levels, testes weight, Leydig cell number, gene and protein expression	PND35	[44]
BDE-209	10	Mouse	Oral gavage	PND21–70	No significant change at the dose of exposure	PND71	[45]
BDE-209	0.025	Mouse	Sub-cutaneous injection	PND1–5	Testis weight, sperm count, elongated spermatids	PND84	[46]
BDE-209	0.025	Mouse	Sub-cutaneous injection	PND1–5	Serum testosterone	PND84	[47]
**Adult studies**							
BDE-47	0.03	Rat	Oral gavage	48 days	Multinucleated giant cells in testis, serum testosterone levels	24 h after exposure	[48]
BDE-47	0.03	Rat	Oral gavage	84 days of exposure	Testosterone levels, organization of the seminiferous epithelium	After 84 days of exposure	[49]
DE-71	3	Rat	Oral gavage	PND90–93	Serum LH, estrone, androstenedione, and testosterone levels	PND93	[50]
BDE-209	0.2	Rat	Oral gavage	PND77–105	Seminal vesicle/coagulation of gland weight	PND105	[51]
BDE-3	1.5	Mouse	Oral gavage	PND105–147	Sperm count	PND148	[52]
BDE-47	0.0015	Mouse	Oral gavage	PND56–86	Sperm capacitation and sperm motility	PND86	[53]
BDE-47	10	Mouse	Oral gavage	PND56–92	Sperm levels in the epididymal lumen	PND93	[54]
BDE-209	7.5	Mouse	Oral gavage	PND42–70	Sperm number, germinal epithelium	PND70	[55]
BDE-209	20	Mouse	Oral gavage	PND28–140	Testicular expression of genes and proteins	PND140	[56]

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
