# Peer review of "Mechanisms of Male Reproductive Toxicity of Polybrominated Diphenyl Ethers"

_ijms, 2022, doi:10.3390/ijms232214229_

Round 1
Reviewer 1 Report
The authors have undertaken a study to explain Mechanisms of Male Reproductive Toxicity of Polybrominated Diphenyl Ethers.
Specific novelty is not observed in the present research. The novelty of the article should be mentioned.
In a review article, new references should be used. It is better to use up-to-date references throughout the article.
It is better to use figures to explain the mechanisms within the article.
Author Response
The authors have undertaken a study to explain Mechanisms of Male Reproductive Toxicity of Polybrominated Diphenyl Ethers. Specific novelty is not observed in the present research. The novelty of the article should be mentioned.
Our manuscript is a review paper, and as such it does not bear any novel experimental findings. The novelty of our review consists in the critical analysis of existing body of literature on male reproductive toxicity of PBDEs in humans and in laboratory experiments to: (1) identify the most sensitive health outcomes and (2) to identify the most plausible molecular mechanisms of PBDE reprotoxicity via the analysis of all previously suggested mechanistic hypotheses and all published evidence in their support. We analyzed 13 groups of hypotheses, and this analysis allowed us to reject many popular hypotheses. Additionally, we demonstrated that hypotheses well supported by high-quality evidence point to the disruption by PBDE of different aspects the same molecular network, illustrated at figure 3. As such we believe that our review contains significant novelty, which was emphasized in conclusions of the study, as well as in Abstract and introduction.
In a review article, new references should be used. It is better to use up-to-date references throughout the article.
Papers were extracted form PubMed using systematic criteria. This search was accomplished in June 2021, and therefore some papers that were published after this date were not covered by our review. To fix this we conducted a new round of search following the same criteria on November 7, 2022. This search retrieved 3 additional papers, which are now included to the review.
It is better to use figures to explain the mechanisms within the article.
We agree with the reviewer that figures are the most efficient way to deliver mechanistic information. However, in traditions of scientific publishing, only figures that deliver novel, original information are accepted by journals. Given that we do not contribute to the understanding of oxidative stress, estrogenic signaling and other hypothetical mechanisms of PBDE toxicity we suppose that corresponding figures will not be welcomed by the journal. Therefore, we limited our graphic information to these molecular mechanisms which result from our study, - see figure 3.
Reviewer 2 Report
The present review is well organized and written, however is needed further improvements. Review talks about the toxicity effects of PBDEs on male reproductive health. Authors demonstrated the effects of PBDEs on sperm development, testicular tissues changes, sperm quality. Its interest to note some studies using developed techniques in understanding the potential adverse effects of PBDEs on sperm epigenetics ad proteomics. Authors clarified the different mechanisms of PBDE such as mitochondrial dysfunction, oxidative stress, inflammation pathways, and steroidogenesis effects. However, I suggested major revision. Some comments should be addressed for publication in this review. I attached all my comments in the PDF file of the manuscript, please authors reasoned and improved your work according to the comments.

Author Response
The present review is well organized and written, however is needed further improvements. Review talks about the toxicity effects of PBDEs on male reproductive health. Authors demonstrated the effects of PBDEs on sperm development, testicular tissues changes, sperm quality. Its interest to note some studies using developed techniques in understanding the potential adverse effects of PBDEs on sperm epigenetics ad proteomics. Authors clarified the different mechanisms of PBDE such as mitochondrial dysfunction, oxidative stress, inflammation pathways, and steroidogenesis effects. However, I suggested major revision. Some comments should be addressed for publication in this review. I attached all my comments in the PDF file of the manuscript, please authors reasoned and improved your work according to the comments.
We are very thankful for the positive review, and we addressed all the edits as recommended in the PDF file.
In lines 38-39 our text was: “Globally, three major commercial compounds of PBDEs have been produced and used: 38 deca-, octa-, and penta- [5].” The reviewer suggested “please, revise it, I think they are tetra-, penta-, hepta-, octa-, and deca-congeners.” Instead, we changed “compounds” to “mixtures” in this sentence. Now it reads: “Globally, three major commercial mixtures of PBDEs have been produced and used: 38 deca-, octa-, and penta- [5].” We are thankful to the reviewer for these comments, and now this sentence is more accurate as there were 3 major industrial mixtures, although each of them was composed of a spectrum of compound (see our reference #5, or PBDE article on the US EPA website).
Round 2
Reviewer 2 Report
No further comments are required. Authors addressed all the comments.